# DROWNING DETECTION BASED ON YOLOv8 IMPROVED BY GP-GAN AUGMENTATION

## ABSTRACT

Drowning is a significant safety issue worldwide, and a robust computer vision-based alert system can easily prevent such tragedies in swimming pools. However, due to domain shift caused by the visual gap (potentially due to lighting, indoor scene change, pool floor color etc.) between the training swimming pool and the test swimming pool, the robustness of such algorithms has been questionable. To address this issue, we propose a domain-aware data augmentation pipeline based on Gaussian Poisson Generative Adversarial Network (GP-GAN). Combined with YOLOv8, we demonstrate that such a domain adaptation technique can significantly improve the model performance (from 0.24 mAP to 0.82 mAP) on new test scenes.

## 1 INTRODUCTION

Drowning is a significant global health problem, accounting for 7% of all injury-related deaths. Each year, an estimated 236,000 people die from drowning (1). To tackle this issue, we aim to utilize computer vision and machine learning techniques. Drowning detection can be simplified into a swimmer detection problem: When a swimmer disappeared within the pool area, there is a non-trivial chance of drowning.

Although previous works have made contributions to the swimmer detection problem (2) (3) (4) (5), they may not be reliable for new domains(due to train/test visual gap). Fortunately, recent advances in domain adaptation have shown promise in addressing the domain shift problem. For example, GANs have been proposed to bridge the domain gap in turbine detection applications (6). Following previous work, we proposes using GP-GAN (7) to address the domain shift issue. We conducted experiments on a swimmer detection training set (8) and a test set (9), which featured water scenes different from each other. We trained GP-GAN to augment the data and found that our domain adaptation approach outperformed YOLOv8 (10) in terms of predictions.

## 2 METHODS

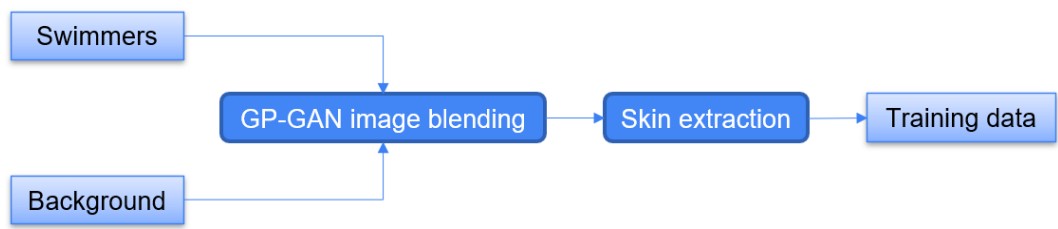

Figure 1: The generating process of images in the augmented training set. Background images of testing domain and swimmer snapshots from training domain are feed into the GP-GAN, followed by a skin extraction algorithm to maintain the original skin color

Most human detection datasets (11) typically feature pedestrians in street scenes, making them unsuitable for training a swimmer detection model. Unfortunately, there are very few available datasets

for swimmer detection (12) (13) (8), and only a handful of them meet our requirement, so we collected a set of 200 appropriate images from (8) to be the original training set. For the test set, we manually labeled 55 images from videos (9) captured by a surveillance camera installed in a swimming pool. They have a different water scene from the training set. We trained a YOLOv8 using the ImageNet pretrained weights as initialization.

In addition to the original training set, we generated an augmented training set by GP-GAN to compare the results. GP-GAN (7) is a powerful tool for blending two images realistically, which can help us blend swimmers into a swimming pool and generate new training data to close the domain gap between different water scenes.

The generating process is shown in Fig.1. We first extract swimmer images from the original dataset and collect several background images of swimming pools that are similar to the water scene of the test set. Next, we blend the swimmer images into the backgrounds using GP-GAN, with random location, rotation and resizing. However, the swimmers in the output images given by GP-GAN may be blurred, so we extracted their skin from the original images and put it back into the output images to ensure that all swimmers are clear enough. We use an established method (14) to extract human skin pixels from a picture: we convert the RGB color space into the YCbCr color space, and the Cb and Cr range distribution is used to determine the global skin color classifier, range as $(75 < Cb < 135)$ and $(130 < Cr < 185)$. We generated 200 images with above process, which constitute our new training set and have a similar domain to the test set.

## 3 RESULTS

Our main results are shown in Fig.2, the model's performance when trained on the original training set was even worse than that of a typical human detection dataset (11) (walker line), with a maximum mAP of just 0.24 and a minimum IOU of 0.5 on the test set. This poor score was mainly due to the significant domain differences between the water scene in the training set and the scene in the test set.

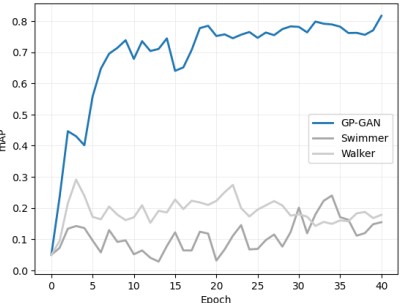

Figure 2: Results of different datasets.    Figure 3: Experiments on the new dataset.

With the help from the augmented training set, we were able to improve the performance of YOLOv8 on the swimmer detection task. After 40 epochs of training, the model achieved the best mAP of 0.82 shown in Fig.2 (GP-GAN line), which is a significant improvement over the original dataset. All training experiments were conducted using the same pretrained model and hyperparameters.

We trained the model six times on the new dataset, and the results with error bars are shown in Fig.3. The model demonstrated consistent and stable performance on the new training set, indicating that the generated data helped to close the domain gap.

## 4 CONCLUSION

We conclude that the GP-GAN is a promising technique that can battle the domain gap between training and testing in the swimming pool drowning detection problem. We believe that with more training data (like drowning ImageNet) to transfer knowledge from, such detection modules can be effectively saving thousands of lives worldwide.

## 5 URM STATEMENT

The authors acknowledge that at least one key author of this work meets the URM criteria of ICLR 2023 Tiny Papers Track.

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

APPENDIX

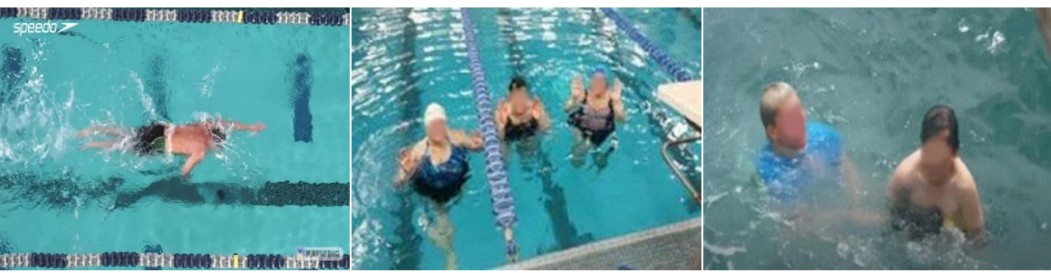

Figure 4: The original training set, which contains 200 images collected from available datasets.

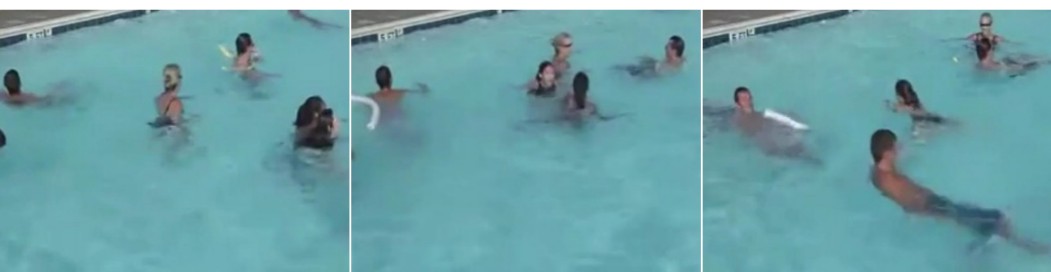

Figure 5: The test set, which contains 55 images extracted from videos caught by a surveillance camera near a swimming pool that has a different water scene from the training set.

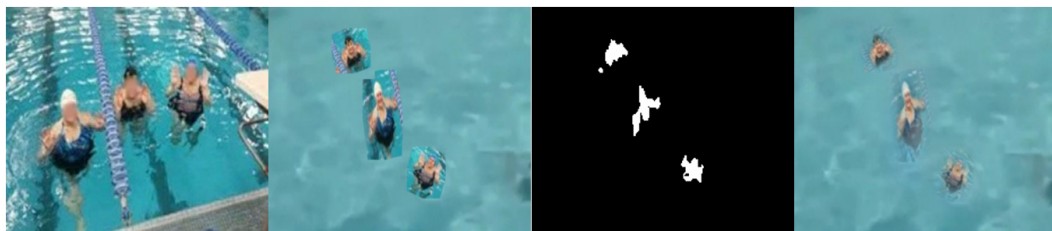

Figure 6: The generating process of the new training set: images from left to right are the original image, the copy-paste image, the extracted skin and the final training image.

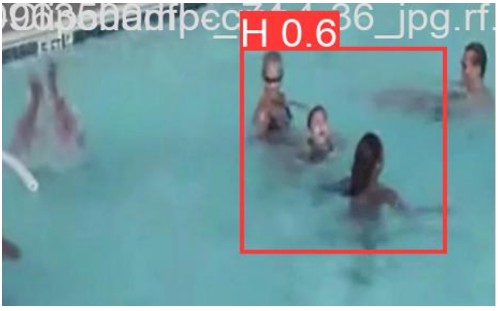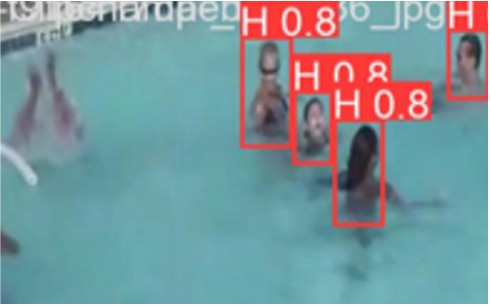

Figure 7: Predictions given by the model trained by the original dataset (left) and the model trained by the augmented dataset (right).

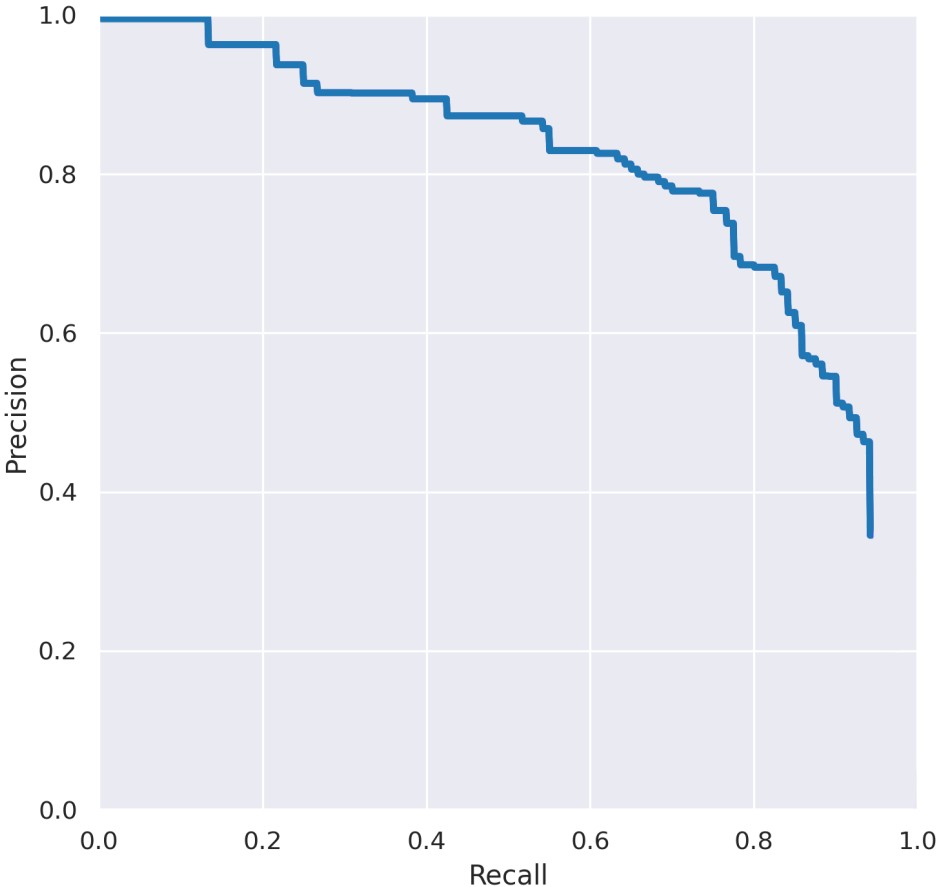

Figure 8: This is the Precision-Recall curve of the YOLOv8 trained by the augmented dataset. Precision measures the accuracy of positive predictions, whereas recall measures the model's completeness in identifying relevant instances. In the swimmer detection problem, recall is more important than precision to minimize the risk of missing any potential danger or drowning incidents.

