# OpenReview forum: "Drowning Detection based on YOLOv8 improved by GP-GAN Augmentation"
_ICLR.cc/2023/TinyPapers — Submitted to Tiny Papers @ ICLR 2023_

### Official Review · Reviewer_4ag1 · 2023-03-30

**Confidence:** 4

**Summary Of Contributions:**

To tackle the problem of domain shift for robust drowning alert system, the authors propose a pipeline based on GP-GAN for data augmentation and YOLOv8. Experimental results are provided to show that this approach significantly increases performance.

**Rating:**

Great Start (GS): a submission which meets some of the reviewing criteria but has room for improvement

**Strengths And Weaknesses:**

Strengths:
- The paper is well written and address a specific problem of domain shift for drowning detection in pools.
- Proper references have been provided for relevant literature.
- While its certainly difficult to cover everything in 2 pages, the authors have done a good job of providing sufficient information.

Weakness:
- Providing links to a repository for code/data would certainly help make the submission more reproducible.

**Suggested Changes:**

Comments:
- Currently the test set contains images from one pool (based on ref. 9). Adding more varied test data from different settings can further support your claim.

---

### Official Review · Reviewer_s7Aw · 2023-03-30

**Confidence:** 4

**Summary Of Contributions:**

This work proposes a method based on GP-GAN to address the domain shift issue.

**Rating:**

Needs Clarification (NC): a submission which does not meet the reviewing criteria and needs clarification for its described problem or solution

**Strengths And Weaknesses:**

Strengths:

1. Proposes a domain-aware data augmentation pipeline for addressing the domain shift issue.

Weaknesses:

1. The contributions are very limited. GP-GAN has been widely investigated in many areas including the data augmentation, e.g., https://arxiv.org/pdf/1703.07195.pdf . Hence, the originality and novelty are very poor.

2. The presentation and written are poor.

3. This paper lacks of technical analysis.

**Suggested Changes:**

1. I strongly suggest the authors to reconsider this idea since this work does not solve an important problem in AI.

2. Refer to the weaknesses part.

---

### Official Review · Reviewer_sFuv · 2023-03-31

**Confidence:** 3

**Summary Of Contributions:**

Review for paper #42

**Rating:**

Great Start (GS): a submission which meets some of the reviewing criteria but has room for improvement

**Strengths And Weaknesses:**

This paper proposes a data augmentation technique for drowning detection through GAN-style generations. The contribution of this paper is clearly presented.

Suggestions:

First, this paper is not well-motivated. The author claims to tackle the visual gap, which is not clearly identified. ''Potentially caused by xxx'' is not adequate to help us identify and verify the source of gap.

Second, this paper tackles the drawing detection task from the aspect of domain adaptation. Based on this, I suggest the author at least compare the proposed method with representative domain adaptation techniques, thereby generally verifying the effectiveness of the proposed method.

Third, the contribution of the proposed designs should be well analyzed in ablations.

**Suggested Changes:**

See above

---

### Meta-Review · Area_Chair_j2fF · 2023-04-08

**Recommendation:** Invite to revise
**Confidence:** 4

**Metareview:**

Based on the three reviews provided, it can be seen that the paper proposes a data augmentation technique for drowning detection through GAN-style generations, using GP-GAN and YOLOv8. Overall, the paper appears to have strengths in its clear presentation of the contribution and well-written language. However, the paper also has several weaknesses, including poor technical analysis, limited originality, poor motivation, and lack of analysis in ablations. Review 3 notes that providing a link to a repository for code and data would be beneficial for reproducibility.


**Summary:**

The main message of the paper is to propose a data augmentation technique using GP-GAN and YOLOv8 for drowning detection in pools to tackle the problem of domain shift.

**Reason For Not Giving A Higher Recommendation:**

As discussed by the reviewers, the submission still has some room for improvement.

**Reason For Not Giving A Lower Recommendation:**

N/A

---

### Decision · Program_Chairs · 2023-04-10

No revision received; not invited to archive